# Self-supervised Visualisation of Microscopy Datasets

## Abstract

Self-supervised learning methods based on data augmentations, such as SimCLR, BYOL, or DINO, allow obtaining semantically meaningful representations of image datasets and are widely used prior to supervised fine-tuning. A recent self-supervised learning method, $t$-SimCNE, uses contrastive learning to directly train a 2D representation suitable for visualisation. When applied to natural image datasets, $t$-SimCNE yields 2D visualisations with semantically meaningful clusters. In this work, we used $t$-SimCNE to visualise medical image datasets, including examples from dermatology, histology, and blood microscopy. We found that increasing the set of data augmentations to include arbitrary rotations improved the results in terms of class separability, compared to data augmentations used for natural images. Our 2D representations show medically relevant structures and can be used to aid data exploration and annotation, improving on common approaches for data visualisation.

## 1 Introduction

Medical image datasets have been quickly growing in size and complexity (Litjens et al., 2017; Topol, 2019; Zhou et al., 2021). Whereas medical professionals can analyse, annotate, and classify individual images, tasks involving large batches of images, ranging from data curation and quality control to exploratory analysis, remain challenging.

Self-supervised learning (SSL) has recently emerged in computer vision as the dominant paradigm for learning image representations suitable for downstream tasks (Balestriero et al., 2023), and it has increasingly been adopted in medical imaging (Huang et al., 2023). In *contrastive learning* methods, such as SimCLR (Chen et al., 2020), BYOL (Grill et al., 2020), or DINO (Caron et al., 2021), data augmentation is used to generate different *views* of each image and a deep network is trained to keep these views close together in the representation space. However, the learned representations are typically high-dimensional.

Recently, Böhm et al. (2023) suggested a self-supervised contrastive method, called $t$-SimCNE, for 2D visualisation of image datasets. Using natural image datasets, the authors demonstrated that $t$-SimCNE obtains semantically meaningful visualisations, representing rich cluster structure and highlighting artefacts in the

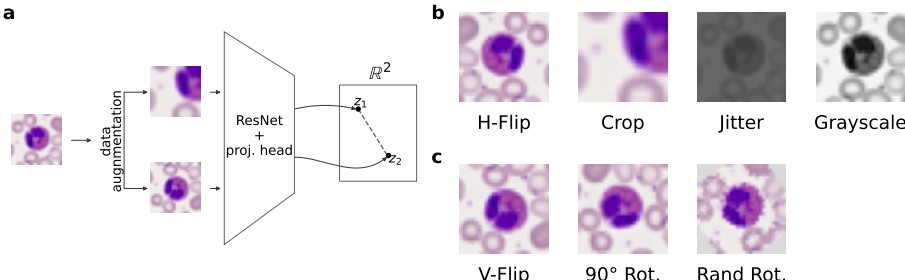

Figure 1: **(a)** In $t$-SimCNE, the network is trained to map two random augmentations of an input image to close locations in the 2D output space. **(b)** Augmentations used for natural images in $t$-SimCNE. **(c)** Additional augmentations suggested here for medical images.

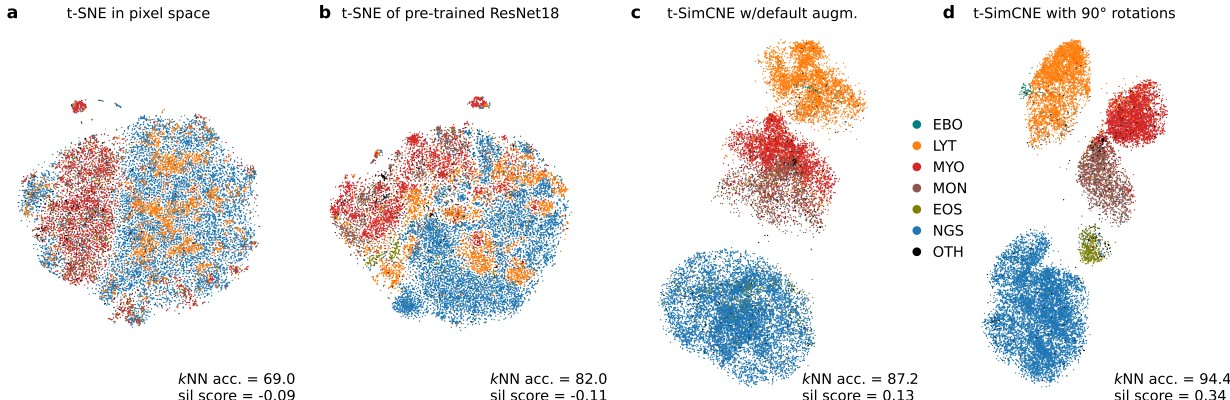

Figure 2: Visualisations of the Leukemia dataset. Small classes shown in black ('OTH' in the legend). $k$NN accuracy and silhouette scores shown in each panel. **(a)** $t$-SNE of the original images in the pixel space. **(b)** $t$-SNE of the 512-dimensional representation obtained via an ImageNet-pretrained ResNet18 network. **(c)** $t$-SimCNE using the same augmentations as in Böhm et al. (2023). **(d)** $t$-SimCNE using augmentations including 90° rotations and flips. Note that the EBO class is well separated here, despite only consisting of 78 images.

data. Their methods clearly outperformed existing 2D embedding methods like $t$-SNE (Van der Maaten & Hinton, 2008) and UMAP (McInnes et al., 2020) for natural image data.

Here we apply $t$-SimCNE to several medical microscopy datasets, and demonstrate that it yields medically relevant visualisations, outperforming $t$-SNE visualisations of pretrained networks. Furthermore, we show that the results improve when using rotational data augmentations (Figure 1a) informed by the rotational invariance of microscropy images. Our code is available at `https://anonymous.4open.science/r/medical-tsimcne-8CF4`.

## 2 Related work

Contrastive learning methods have been widely applied to medical image datasets (Huang et al., 2023) but usually as pre-training for downstream tasks such as classification or segmentation. Some recent works visualised high-dimensional SSL representations; e.g. Cisternino et al. (2023) used UMAP of DINO to visualise histopathology data. In contrast, our focus is on self-supervised visualisations trained end-to-end.

Contrastive learning relies on data augmentations to create several views of each image, and the choice of data augmentations plays a crucial role in methods' success (Tian et al., 2020). A large number of works explored data augmentations for medical images in a supervised setting (Chlap et al., 2021; Goceri, 2023). In the self-supervised context, van der Sluijs et al. (2023) studied the effect of augmentations on the representation of X-ray images. For histopathology images, Kang et al. (2023) advocated for using rotations and vertical flips, as well as staining-informed colour transformations, while some other works used neighbouring patches as positive pairs (Li et al., 2021; Wang et al., 2021).

## 3 Background: SimCLR and $t$-SimCNE

SimCLR (Chen et al., 2020) produces two augmentations for each image in a given mini-batch of size $b$, resulting in $2b$ augmented images. Each pair of augmentations forms a so-called *positive pair*, whereas all other possible pairs in the mini-batch form *negative pairs*. The model is trained to maximise the similarity between the positive pair elements while simultaneously minimising the similarity between the negative pair elements.

Table 1: Summary of the datasets used for our evaluation.

| Dataset | Image dim. | Sample size | Classes | Ref. |
|---------|-----------|-------------|---------|------|
| Leukemia | $28 \times 28$ | 18 365 | 7 | Matek et al. (2019a) |
| BloodMNIST | $28 \times 28$ | 17 092 | 8 | Yang et al. (2023) |
| DermaMNIST | $28 \times 28$ | 10 015 | 2 | Yang et al. (2023) |
| PathMNIST | $28 \times 28$ | 107 180 | 9 | Yang et al. (2023) |
| PCam16 | $96 \times 96$ | 327 680 | 2 | Veeling et al. (2018) |

An augmented image $x_i$ is passed through a ResNet (He et al., 2016) *backbone* (or any suitable backbone) to give the latent representation $h_i$, which is then passed through a fully-connected *projection head* with one hidden layer to yield the final output $z_i$. SimCLR employs the InfoNCE loss function (van den Oord et al., 2019), which for one positive pair $(i, j)$ can be written as

$$\ell_{ij} = -\log \frac{\exp\big(\mathrm{sim}(z_i, z_j)/\tau\big)}{\sum_{k \neq i}^{2b} \exp\big(\mathrm{sim}(z_i, z_k)/\tau\big)} \,, \tag{1}$$

where $\mathrm{sim}(x, y) = x^\top y / \big(\|x\| \cdot \|y\|\big)$ is the cosine similarity and $\tau$ is a hyperparameter that was set to $1/2$ in Chen et al. (2020). Even though the loss function operates on $z_i$ (typically 128-dimensional), for downstream tasks, SimCLR uses the representations $h_i$ (Bordes et al., 2022), typically at least 512-dimensional.

The idea of $t$-SimCNE (Böhm et al., 2023) is to make the network output ($z_i$) two-dimensional so that it is directly suitable for data visualisation. $t$-SimCNE replaces the scaled cosine similarity used in Chen et al. (2020) with the Cauchy similarity function $(1 + \|x - y\|^2)^{-1}$ as in $t$-SNE (Van der Maaten & Hinton, 2008) to ensure that the embeddings are not constrained in a circle. The resulting loss function is

$$\ell_{ij} = -\log \frac{1}{1 + \|z_i - z_j\|^2} + \log \sum_{k \neq i}^{2b} \frac{1}{1 + \|z_i - z_k\|^2} \,. \tag{2}$$

Böhm et al. (2023) found that directly optimizing this loss is difficult, and suggested a three-stage process. The first stage (1000 epochs) used a 128-dimensional output which was then replaced with a 2D output and fine-tuned in the subsequent two stages (500 epochs). In the first stage, the output dimension is set to 128, and the network is trained for 1000 epochs. In the second stage, the 128-dimensional output layer is removed and replaced with a randomly initialized 2-dimensional output layer, which is fine-tuned for 50 epochs while the rest of the network is frozen. In the third stage, the entire network is unfrozen and trained for further 450 epochs. For their experiments on CIFAR datasets, the authors used a ResNet18 with a modified first layer kernel size of $3 \times 3$, and a projection head with hidden layer size of 1024 (Figure 1a).

## 4 Experimental setup

**Datasets**  We used five publicly available medical image datasets with sample sizes ranging from 10 000 to over 300 000 (Table 1). Three datasets were taken from the MedMNISTv2 collection (Yang et al., 2023), all consisting of $28 \times 28$ RGB images. DermaMNIST with 7 classes is based on the HAM10000 dataset (Tschandl et al., 2018), a collection of multi-source dermatoscopic images of common pigmented skin lesions with 7 classes, which we reduced to 2 labels: melanocytic nevi and other skin conditions. BloodMNIST is based on a dataset of microscopy images of individual blood cells from healthy donors (Acevedo et al., 2020), with 8 classes corresponding to cell types. PathMNIST is based on a dataset of non-overlapping patches from colorectal cancer histology slides (Kather et al., 2019), categorized into 9 classes corresponding to tissue types. The Leukemia dataset (Matek et al., 2019b) contains microscopy images of white blood cells taken from patients, some of which were diagnosed with acute myeloid leukemia. We resized $224 \times 224$ images to $28 \times 28$ and merged 9 rare classes ($< 80$ cells) into one, obtaining 7 classes. The Patch Camelyon16 (PCam16) dataset (Veeling et al., 2018), adapted from the Camelyon16 challenge (Bejnordi et al., 2017), consists of

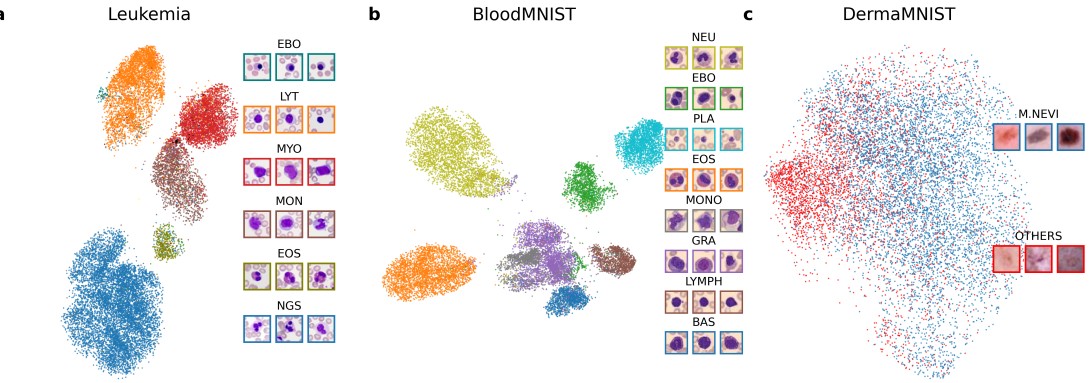

Figure 3: **(a)** *t*-SimCNE visualisation of the Leukemia dataset. Only a subset of classes is listed in the legend. **(b)** *t*-SimCNE visualisation of the Bloodᴍɴɪsᴛ dataset. **(c)** *t*-SimCNE visualisation of the Dermaᴍɴɪsᴛ dataset. In all three cases, we used augmentations including 90° rotations and vertical flips.

$96 \times 96$ patches from breast cancer histology slides with two classes: metastases and non-metastases. A patch was classified as metastases if there was any amount of tumor tissue in its central $32 \times 32$ region.

**Augmentations** Böhm et al. (2023) worked with natural images and used the same data augmentations as Chen et al. (2020): cropping, horizontal flipping, color jittering, and grayscaling (Figure 1b). Here we used all of these augmentations with the same hyperparameters and probabilities (see Table 4 for ablations). We reasoned that the semantics of microscopy or pathology images should be invariant to arbitrary rotations and arbitrary flips (Kang et al., 2023). For that reason we considered two additional sets of augmentations: (i) vertical flips and arbitrary 90° rotations which were applied with a probability of 50%; (ii) on top of that, rotations by any arbitrary angle within the range from $-45$ to 45 degrees applied with probability 100% (Figure 1c). In each case, all possible rotations were equally likely. When rotating an image by an angle that is not a multiple of 90°, the corners need to be filled in (Figure 1c, right). For this we used the average color of all border pixels across all images in a given dataset. This color was dataset specific, but the same for all images in a dataset.

**Architecture and training** We used the original *t*-SimCNE implementation (Böhm et al., 2023) with default parameters unless stated otherwise. For PCam16, we used the unmodified ResNet18 (He et al., 2016) without the fully-connected layer. All networks were trained from scratch on an NVIDIA RTX A6000 GPU with the batch size of 1024, except for PCam16 where we had to reduce the batch size to 512 to fit it into GPU memory.

**Baselines** For comparison, we applied *t*-SNE to images in pixel space, in pretrained ResNet representation, and in SimCLR representation. The SimCLR models had the same architecture as *t*-SimCNE models but with 128D output and were trained with SimCLR loss (Eq. 1) for 1000 epochs. We then applied *t*-SNE to the 512-dimensional representation before the projector head. As an alternative SSL baseline, we used BYOL Grill et al. (2020). Unlike SimCLR and *t*-SimCNE, BYOL does not use negative pairs but rather a student-teacher architecture where the student network is trained to predict the representations produced by the teacher network, and the parameters of the teacher network are updated as a slow-moving average of the student. We trained the BYOL model for 1000 epochs as well, and applied *t*-SNE to the final 512D representation. We took ImageNet-pretrained ResNet18 and ResNet152 models from the PyTorch library. We resized all images to $256 \times 256$, center cropped to $224 \times 224$, and normalized, following (He et al., 2016). The resulting representations were 512D and 2048D respectively. We used openTSNE 1.0.1 (Poličar et al., 2024) with default settings to reduce to 2D. When doing *t*-SNE of the PCam16 data in pixel space, we first performed principal component analysis and only used the first 100 PCs as input to *t*-SNE.

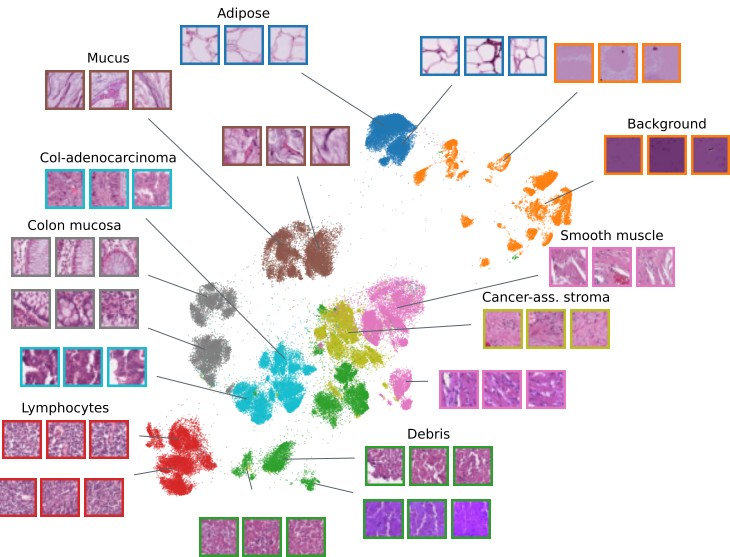

Figure 4: *t*-SimCNE visualisation of the PathMNIST dataset. Colours correspond to classes. Images correspond to three random points close to the tip of the annotation line.

**Evaluation in 2D** Our focus is on low dimensional visualisation of medical image datasets. Hence, we evaluated the quality of the 2D embeddings on two downstream tasks: classification and clustering. We used *k*NN classification accuracy (Cover & Hart, 1967; Fix & Hodges, 1989) (with $k = 15$ and a 9:1 training/test split) to measure how *well* the classes are separated from each other, while the silhouette score (Rousseeuw, 1987) measures how *far* they are separated from each other. Silhouette scores range from -1 to 1, calculated as $(b−w)/\max(w, b)$, where $w$ is the average intra-class distance and $b$ is the average nearest-cluster distance. A high silhouette score indicates clear class distinction, complementing the *k*NN accuracy metric. We used the implementation provided by the Scikit-learn library (Pedregosa et al., 2011).

**Evaluation for SimCLR** To evaluate SimCLR representations in 512D, we used linear evaluation, which is the standard approach in the self-supervised learning literature. Here, we trained SimCLR on the training and validation sets (put together). Subsequently, we used the same data to train a linear classifier via `sklearn.linear_model.LogisticRegression(solver="saga", penalty=None)` after scaling the representations with a `sklearn.preprocessing.StandardScaler()`. Finally, we evaluated classifier performance on the test set. Note that the test set was used neither for self-supervised pre-training nor for the supervised training parts in these experiments.

## 5    Results

In this study, we asked (i) how the contrastive visualisation method *t*-SimCNE (Böhm et al., 2023) could be applied to medical image datasets, and (ii) if the set of data augmentations could be enriched compared to what is typically used on natural images.

Firstly, we considered the Leukemia dataset (Figure 2). Naive application of *t*-SNE to the images in pixel space resulted in an embedding with little class separation and low *k*NN accuracy of 67.4% (Figure 2a). Using an ImageNet-pretrained ResNet and then embedding them with *t*-SNE improved the *k*NN accuracy to 82.2% but visually had poorly separated classes (Figure 2b). Training *t*-SimCNE with default data augmentations gave embeddings with 86.7% *k*NN accuracy (Table 2) and much better visual class separation (Figure 2c and Table 3). This shows that *t*-SimCNE can produce meaningful 2D visualisations of medical image datasets.

Since the semantics of blood microscopy images are rotationally invariant, we included 90° rotations and vertical flips into the set of data augmentations. When training *t*-SimCNE with this set of data augmentations,

Table 2: The $k$NN accuracy of 2D embeddings. Means $\pm$ standard deviations over three runs; PCam16 experiments had only one run due to its large size. Bold numbers correspond to the best performance of $t$-SimCNE across different augmentation strategies. DermaMNIST showed poor performance overall and hence nothing is highlighted.

| Method | | Dataset | | | | |
|---|---|---|---|---|---|---|
| | | Leukemia | BloodMNIST | DermaMNIST | PathMNIST | PCam16 |
| $t$-SimCNE | def. augm. | $86.3 \pm 0.7$ | $90.4 \pm 0.3$ | $77.3 \pm 0.6$ | $97.2 \pm 0.2$ | $92.6$ |
| | + 90° rot. | $94.4 \pm 0.1$ | $\mathbf{93.0 \pm 0.3}$ | $77.5 \pm 0.3$ | $\mathbf{98.0 \pm 0.0}$ | $\mathbf{93.1}$ |
| | + rand. rot. | $\mathbf{95.1 \pm 0.2}$ | $92.9 \pm 0.1$ | $80.1 \pm 0.7$ | $97.3 \pm 0.0$ | $90.8$ |
| $t$-SNE of SimCLR | def. augm. | $95.0 \pm 0.1$ | $94.0 \pm 0.1$ | $81.9 \pm 0.1$ | $98.1 \pm 0.0$ | $96.3$ |
| | + 90° rot. | $95.9 \pm 0.1$ | $95.8 \pm 0.1$ | $80.8 \pm 0.6$ | $98.4 \pm 0.0$ | $96.4$ |
| | + rand. rot. | $95.6 \pm 0.1$ | $95.4 \pm 0.1$ | $82.2 \pm 0.2$ | $97.9 \pm 0.1$ | $94.9$ |
| $t$-SNE of BYOL | def augm. | $93.1 \pm 0.4$ | $90.6 \pm 0.0$ | $80.1 \pm 0.9$ | $93.0 \pm 1.0$ | |
| $t$-SNE | pixel space | $69.0$ | $73.2$ | $78.0$ | $56.9$ | $76.9$ |
| | ResNet18 | $82.0$ | $78.1$ | $81.9$ | $87.2$ | $86.7$ |
| | ResNet152 | $72.9$ | $72.9$ | $81.0$ | $88.8$ | $86.4$ |

Table 3: Silhouette scores (Section 4) of 2D embeddings. Same format as in Table 2.

| Method | | Dataset | | | | |
|---|---|---|---|---|---|---|
| | | Leukemia | BloodMNIST | DermaMNIST | PathMNIST | PCam16 |
| $t$-SimCNE | def. augm. | $0.13 \pm 0.00$ | $0.40 \pm 0.00$ | $0.13 \pm 0.01$ | $0.45 \pm 0.02$ | $0.04$ |
| | + 90° rot. | $0.33 \pm 0.01$ | $0.44 \pm 0.03$ | $0.11 \pm 0.00$ | $\mathbf{0.48 \pm 0.06}$ | $\mathbf{0.05}$ |
| | + rand. rot. | $\mathbf{0.52 \pm 0.02}$ | $\mathbf{0.50 \pm 0.01}$ | $0.13 \pm 0.06$ | $0.41 \pm 0.03$ | $0.05$ |
| $t$-SNE of SimCLR | def. augm. | $0.21 \pm 0.01$ | $0.37 \pm 0.00$ | $0.14 \pm 0.00$ | $0.23 \pm 0.01$ | $0.16$ |
| | + 90° rot. | $0.23 \pm 0.01$ | $0.35 \pm 0.02$ | $0.14 \pm 0.01$ | $0.25 \pm 0.01$ | $0.13$ |
| | + rand. rot. | $0.21 \pm 0.00$ | $0.37 \pm 0.02$ | $0.16 \pm 0.00$ | $0.26 \pm 0.00$ | $0.06$ |
| $t$-SNE of BYOL | def aug. | $0.16 \pm 0.01$ | $0.32 \pm 0.0$ | $0.13 \pm 0.00$ | $0.19 \pm 0.03$ | |
| $t$-SNE | pixel space | $-0.09$ | $0.07$ | $0.08$ | $-0.05$ | $0.02$ |
| | ResNet18 | $-0.11$ | $0.13$ | $0.14$ | $0.17$ | $0.04$ |
| | ResNet152 | $-0.15$ | $0.03$ | $0.14$ | $0.19$ | $0.05$ |

the $k$NN accuracy increased to 94.4%. Additionally including all possible rotations by an arbitrary angle as data augmentations yielded the highest $k$NN accuracy (95.1%) and the highest silhouette score (0.52), indicating that domain-specific augmentations can further improve $t$-SimCNE embeddings.

We saw three different outcomes across the datasets. On microscropy datasets (Leukemia and BloodMNIST), $t$-SimCNE with random rotations performed the best: it had by far the best silhouette score (Table 3) and visually the most separated classes (Figure 3a,b). SimCLR followed by $t$-SNE has also benefited from rotational augmentations. Compared to $t$-SimCNE, it had slightly higher $k$NN accuracies (Table 2), but much lower silhouette scores (Table 3). The same was true for BYOL followed by $t$-SNE.

On pathology datasets (PathMNIST and PCam16), $t$-SimCNE with 90° rotations performed the best. On PathMNIST (Figure 4), it had the highest silhouette score (Table 3). On PCam16, $t$-SimCNE showed clearer structures compared to SimCLR + $t$-SNE, but this difference was not captured by the silhouette scores which on this dataset were all close to zero (Table 3). This is because it only had two classes, whereas $t$-SimCNE separated images not only by class but also by tissue types (Figure 4); this led to large within-class distances and hence misleadingly low silhouette scores. Finally, on the dermatology dataset (DermaMNIST),

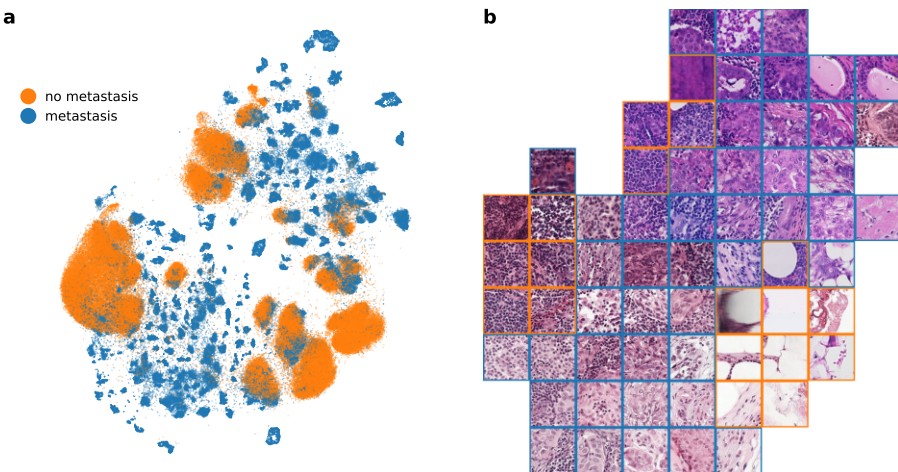

Figure 5: **(a)** *t*-SimCNE visualisation of the PCam16 dataset. **(b)** We superimposed a $10 \times 10$ grid over the embedding and selected one image in each square. Frame colours show image classes. If a square had fewer than 100 images, no image was shown.

performance of all methods was similarly poor: SimCLR and *t*-SimCNE resulted in embeddings not very different from *t*-SNE in pixel space (Figure 3c).

In the pathology datasets, *t*-SimCNE revealed meaningful subclass structures. For instance, in PathMNIST, the *debris* class divided into three unique subsets, with one displaying notably different staining color (Figure 4). In the PCam16 dataset, the embedding distinctly separated patches by the presence of metastasis, influenced by chromatin density and cell size variations. The visual differences in shades of violet between top-right and bottom-left likely indicate a technical artefact from varying staining durations.

To further demonstrate practical usefulness of *t*-SimCNE, we applied it to datasets with data quality issues which often come in the form of duplicates and artefacts. For this, we modified one of the datasets, BloodMNIST, in two different ways, and re-trained the *t*-SimCNE model on each modified dataset. In one experiment, we added 100 duplicates of one image which had been generated by randomly cropping the original image, adding Gaussian noise and jittering the colours. In another experiment, we randomly selected 50 images from the dataset and added black artefacts to each image. Even though in both cases, our perturbations affected only 0.6% and 0.3% of the samples, we observed that all perturbed samples were clustered together in the *t*-SimCNE embedding and could be easily spotted (Figure 6). This is in line with the cluster of duplicate car images that Böhm et al. (2023) observed in their CIFAR-100 embedding.

**Additional experiments** As a control experiment, we applied *t*-SimCNE with 90° rotations and vertical flips to the CIFAR-10 dataset (Krizhevsky et al., 2009). It decreased the *k*NN accuracy from 89% to 76%. This confirms that rotation augmentations are hurtful for natural images since they are not invariant to rotations, unlike microscopy and pathology images.

Even though our focus in this work was on rotational augmentations, we performed an ablation study to check the importance of other standard augmentations. We found that cropping and colour transformations played a key role in the good performance of *t*-SimCNE (Table 4). Without including these augmentations, the performance was low.

Finally, we confirmed that our additional augmentations were helpful for SimCLR when doing standard linear evaluation in 512D (Table 5).

Table 4: Ablation study, removing individual augmentations from $t$-SimCNE. The full set of augmentations included the default $t$-SimCNE augmentations plus arbitrary rotations ($k$NN accuracy is given in percent).

| Augmentations | Leukemia | | BloodMNIST | | PathMNIST | |
|---|---|---|---|---|---|---|
| | $k$NN acc. | Silhouette | $k$NN acc. | Silhouette | $k$NN acc. | Silhouette |
| All | $95.1 \pm 0.2$ | $0.52 \pm 0.02$ | $92.9 \pm 0.1$ | $0.50 \pm 0.01$ | $97.3 \pm 0.0$ | $0.41 \pm 0.03$ |
| No crops | $79.7 \pm 0.6$ | $0.14 \pm 0.00$ | $76.0 \pm 1.1$ | $0.20 \pm 0.01$ | $59.8 \pm 1.1$ | $-0.02 \pm 0.03$ |
| No color jitter | $82.0 \pm 0.2$ | $-0.01 \pm 0.01$ | $90.0 \pm 0.1$ | $0.45 \pm 0.02$ | $94.3 \pm 0.3$ | $0.24 \pm 0.02$ |
| No grayscaling | $95.6 \pm 0.4$ | $0.52 \pm 0.02$ | $92.1 \pm 0.3$ | $0.44 \pm 0.01$ | $98.5 \pm 0.0$ | $0.39 \pm 0.05$ |

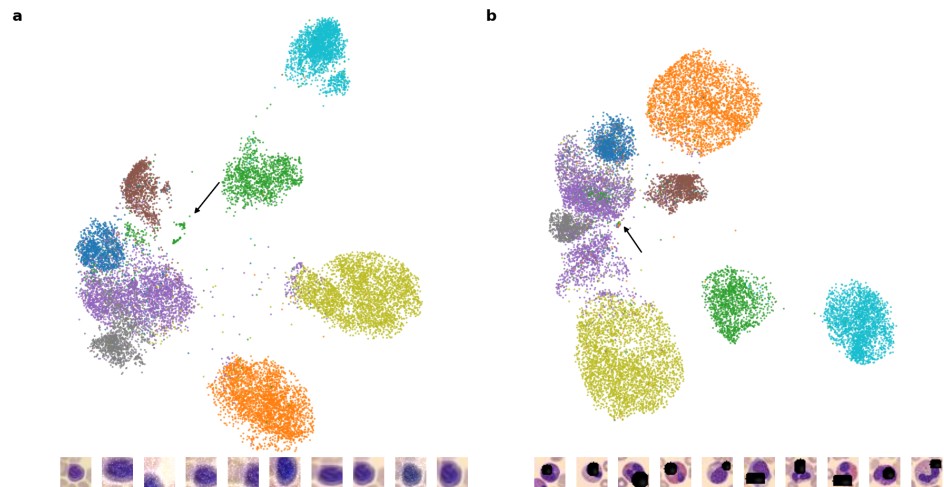

Figure 6: $t$-SimCNE visualisations of the modified BloodMNIST dataset. **(a)** With added 100 duplicates of a single image, with random perturbations. **(b)** With artefacts added to 50 images. In both cases, ten exemplary perturbed images are shown below.

## 6 Discussion

In this paper, we showed that $t$-SimCNE (Böhm et al., 2023) can be successfully applied to medical image datasets, yielding semantically meaningful visualisations, and benefits from rotational data augmentations, leveraging rotational invariance of microscropy images. In agreement with (Böhm et al., 2023), $t$-SimCNE performed better than SimCLR + $t$-SNE combination. Even though SimCLR tended to have slightly higher $k$NN accuracy, the silhouette score was typically much lower: $t$-SimCNE achieved visually much stronger cluster separation, which is useful for practical visualisations. Furthermore, parametric nature of $t$-SimCNE allows to embed new (out-of-sample) images into an existing embedding.

We found that blood microscopy datasets benefited the most from random rotations, while pathology datasets showed the best results with 90° rotations and flips. We believe it is because in blood microscopy images, the semantically meaningful part is always in the center (Figure 3a,b) and so the corners of the image may not be important. In contrast, in histopathology images, the edges of the image may contain relevant information, which may get rotated out of the image and replaced by solid-color triangles (Figure 1c).

One of the datasets, DermaMNIST, exhibited poor results with all analysis methods. This is in agreement with the results of supervised classification reported in the literature: the MedMNIST v2 paper Yang et al. (2023) reported 76.8% classification accuracy on DermaMNIST, which is close to the $k$NN accuracy in the embedding space that we obtained in our experiments. This suggests poor class separability in this dataset,

Table 5: Linear evaluation of SimCLR representations with different augmentations. Test set accuracy.

| Augmentations | Dataset | | |
|---|---|---|---|
| | BloodMNIST | DermaMNIST | PathMNIST |
| def. augm. | $94.5 \pm 0.0\%$ | $82.3 \pm 0.3\%$ | $91.3 \pm 0.3\%$ |
| + 90° rot. | $\mathbf{96.5 \pm 0.2\%}$ | $83.6 \pm 0.5\%$ | $91.9 \pm 0.0\%$ |
| + rand. rot. | $\mathbf{96.5 \pm 0.1\%}$ | $\mathbf{84.5 \pm 0.4\%}$ | $\mathbf{92.7 \pm 0.2\%}$ |

possibly because the images in this dataset are too small to convey medically relevant information. Note that the majority class in DermaMNIST takes 67% of samples.

In conclusion, we argue that $t$-SimCNE is a promising tool for visualisation of medical image datasets. It can be useful for quality control, highlighting artefacts and problems in the data (Figure 6). It can also create a 2D map of cell types, tissue types, or medical conditions, which can be useful not only for clinical purposes but also education and research, potentially combined with an interactive image exploration tool. In the future, it may be interesting to extend $t$-SimCNE to learn representations invariant to technical (e.g. staining) artefacts.

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
