# OpenReview forum: "Self-supervised Visualisation of Microscopy Datasets"
_TMLR — Rejected by TMLR_

### Review · Reviewer_P8ZB · 2024-09-19

**Summary Of Contributions:**

This paper studies and demonstrates the potential to use t-SimCNE for medical image dataset clustering with the aim to provide better dataset image visualization for medical professionals. Given that t-SimCNE is a self-supervised learning model, it has great potential to apply to large medical image dataset without additional input from medical professionals with the domain knowledge. Additionally, this paper studies the impact of commonly used data augmentation methods on the t-SimCNE performance on different medical image dataset. Finally, the experiments show that the clustering result from t-SimCNE reveals meaningful subclass structures, where images with the same class can be further separate into different class, where each class has different medical meanings.

**Audience:**

Yes

**Broader Impact Concerns:**

there are no concerns on the ethical implications.

**Claims And Evidence:**

No

**Requested Changes:**

1. provide additional baselines, especially other machine learning based self-supervised learning models.
2. add more different image augmentation methods and report their impact on clustering
3. introduce more complex and difficult dataset, with higher image size, complex scene with more than one object and other medical images other than cells.
4. clarify how the distance between two images are computed for eval metrics . And justify why eval metrics based on the such distance is a good representation of clustering quality.

**Strengths And Weaknesses:**

strengths:
1. t-SimCNE is tested on wide range of medical dataset, and is shown to have good performance on all dataset
2. The idea of using self-supervised learning for auto-clustering of medical dataset is a good idea. Since it is not relied on domain knowledge and human feedback, clustering can be done at scale and could potentially discover new medical findings.
3. The study of different data augmentations on clustering performance can provide guidance for other researcher when applying clustering using t-SimCNE.

weakness
1. lack of baseline method,   only t-SNE and variant of it is used as baseline, which cannot support the claim that t-SimCNE is a fit for medical image clustering.
2. evaluation metric is questionable. for knn accuracy and silhouette score, both of them are heavily rely on the distance calculation between data points. For 2D images, how to measure such distance is not clearly stated in the paper, nor there is a standard approach for that. Since images with large raw different could still have same semantic meaning, I don't think the proposed eval metric can reflect the actual clustering quality.
3. The datasets used are very similar, all 5 datasets used for medical images of different cells. Although there are difference in terms of cell appearance, it is still very similar and simple task for cluster compared to common computer vision tasks.
4. The augmentation studied is very limited, in the ablation study, only crop, color jitter and grayscale are tested for their impact on clustering performance.
5. The image in each dataset  is small, most dataset consists of image for one cell with size of 28x28, which makes the clustering relatively easy as the image dimension is small and target object is already isolated out.

---

> ### Author Response · Authors · 2024-10-25
> **Response to Reviewer P8ZB**
>
> We thank the reviewer for the thorough feedback and for highlighting the value of our approach. We would like to respond to the following points:
>
> >  lack of baseline method, only t-SNE and variant of it is used as baseline, which cannot support the claim that t-SimCNE is a fit for medical image clustering.
> > [...]
> > provide additional baselines, especially other machine learning based self-supervised learning models.
>
> Please note that we are not aware of other self-supervised methods for end-to-end training of 2D embeddings. On one hand, there are 2D embedding methods like PCA, MDS, t-SNE, UMAP, PHATE, etc. that can be applied to images in pixel space. All these methods are guaranteed to fail when applied to images, because Euclidean distance in pixel space is not meaningful for images. In the manuscript, we used t-SNE in pixel space only as a naive baseline, but it is a priori clear that none of these methods can work well here.
>
> The reviewer probably meant other self-supervised learning methods beyond SimCLR. Indeed, there are many such methods (DINO, BYOL, SimSiam, VICReg, etc.) and they typically all perform similar to each other, at least on the datasets of our size. However, all of them construct high-dimensional embeddings, e.g. 128-dimensional or even higher, and are not suited for 2D embeddings. In the original manuscript, we used SimCLR followed by t-SNE as a baseline, because SimCLR is the method most similar to t-SimCNE, so we felt that SimCLR + t-SNE provides the most meaningful comparison.
>
> We have now added another baseline method from the self-supervised literature, BYOL, also followed by t-SNE. As expected, the resulting performance was quite similar to SimCLR + t-SNE, as we now show in the new rows of Tables 2-3.
>
> > evaluation metric is questionable. for knn accuracy and silhouette score, both of them are heavily rely on the distance calculation between data points. For 2D images, how to measure such distance is not clearly stated in the paper, nor there is a standard approach for that. Since images with large raw different could still have same semantic meaning, I don't think the proposed eval metric can reflect the actual clustering quality
> > [...]
> > clarify how the distance between two images are computed for eval metrics . And justify why eval metrics based on the such distance is a good representation of clustering quality.
>
> There is some misunderstanding here. Both our metrics, kNN accuracy and Silhouette score, were computed in the 2D embedding space, based on the pairwise Euclidean distances between the points. We never compute any metrics in the pixel space, which, as the Reviewer rightly says, would not make much sense.
>
> This is precisely what “metric learning” methods such as t-SimCNE aim to achieve: we have a non-metric space as input (the image pixel space) and we get a metric Euclidean embedding space as output, where the Euclidean distance is meaningful. The kNN accuracy and the silhouette score, both rely on class labels, which were available in all datasets used in our manuscript, but which we never used during training.
>
> > The datasets used are very similar, all 5 datasets used for medical images of different cells. Although there are difference in terms of cell appearance, it is still very similar and simple task for cluster compared to common computer vision tasks.
> > [...]
> > introduce more complex and difficult dataset, with higher image size, complex scene with more than one object and other medical images other than cells.
>
> We admit this limitation and have now adapted the title of our paper such that it better reflects the focus of our work (as was suggested by another Reviewer).  We would like to point out that the PCam16 dataset consists of 96x96 images which are not very small and comparable in size to ImageNet images. We would like to extend our work to even larger datasets and also beyond microscopy images but will leave that for future work.
>
> > The augmentation studied is very limited, in the ablation study, only crop, color jitter and grayscale are tested for their impact on clustering performance.
> > [...]
> > add more different image augmentation methods and report their impact on clustering
>
> To clarify, we have implemented *all* standard augmentations commonly used in the self-supervised learning literature, and added only rotations which have theoretical motivation (the semantic meaning of microscopy images is invariant to rotations).

---

### Review · Reviewer_gYGz · 2024-09-24

**Summary Of Contributions:**

This work applies t-SimCNE, a self-supervised learning method, to visualize medical image datasets such as dermatology, histology, and blood microscopy. By incorporating arbitrary rotations as data augmentations, the 2D representations showed improved class separability and revealed medically relevant structures, enhancing data exploration and annotation.

**Audience:**

No

**Claims And Evidence:**

No

**Requested Changes:**

I think adding more exploration and examples for this claim  'It can be useful for quality control, highlighting artefacts and problems in the data' would strengthen the paper a lot.

**Strengths And Weaknesses:**

This paper conducted a very simple experiment. Basically applied some already known algorithm on medical images with some transform functions in torch transform.

Strength:
- The design of ablation is good, and some datasets are well explored.

weakness:
- The findings about 90 degree augmentation seems very empirical. Is there any theoretical explanations? Since the augmentations applied to input by chance, is there any control on that? Say 100% to 50% chance for example.

---

> ### Author Response · Authors · 2024-10-25
> **Response to Reviewer gYGz**
>
> Thank you for the review and for appreciating our ablation experiments and dataset exploration. We would like to respond to the following points:
>
> > The findings about 90 degrees augmentation seem very empirical. Is there any theoretical explanations?
>
> Here we would like to clarify that our approach is *not* empirical, and in fact is motivated a priori. We know that the microscopy images do not have a preferred orientation; they can be freely rotated and reflected without changing the image semantics. This is in contrast to natural images which typically can only be reflected horizontally.  By using 90 degree rotations, as well as vertical and horizontal flips we train the network to be equivariant to all symmetries of the dihedral group of order 8.
>
> > Since the augmentations applied to input by chance, is there any control on that? Say 100% to 50% chance for example.
>
> In self-supervised learning, all augmentations are applied with some probabilities (that play the role of hyper-parameters). For all augmentations used in SimCLR / t-SimCNE by default, we used the default probabilities. Vertical flips and 90 degree rotations were applied with probability 50% each. Arbitrary rotations were applied within the range from -45 to  45 degrees with probability 100%. We now clarified this in the “Augmentations” paragraph of section 4.
>
> > I think adding more exploration and examples for this claim 'It can be useful for quality control, highlighting artefacts and problems in the data' would strengthen the paper a lot.
>
> We thank the reviewer for raising this issue. To address this, we have now added two additional experiments shown as Fig. 6a and 6b in the revised manuscript. In the first experiment, we introduced 100 duplicate images with small variations into the dataset, and trained the t-SimCNE embedding from scratch. In the resulting embedding, the duplicate images clustered together and were easy to detect (Fig. 6a). In the second experiment,  we introduced artefacts (black pixels) into 50 images. This had a similar effect: the visualization put the modified images into an artefact cluster. We believe that these additional experiments demonstrate that the 2D embeddings via t-SimCNE can be useful for quality control.

---

### Review · Reviewer_YPGk · 2024-09-27

**Summary Of Contributions:**

This paper presents an adaptation of t-SimCNE, a recent self-supervised learning (SSL) method for learning 2-dimensional (2D) representations of natural images, for visualizing medical image datasets. It adds 90-degree rotation augmentation, and random rotation to a set of existing augmentations used by t-SimCNE to model rotation invariance in histopathological and dermatology datasets, namely: Leukemia, BloodMNIST,  DermaMNIST,  PathMNIST, and PCam16. They evaluate the performance of the proposed adaptation on KNN-classification accuracy and Silhouette scores metrics.

**Contributions:** Overall, the paper shows the application of an existing t-SimCNE on histopathological and dermatology datasets, which can be used for informative 2D visualization of the aforementioned datasets. The proposed random-rotation augmentation seems to improve performance for certain datasets.

**Audience:**

Yes

**Broader Impact Concerns:**

No ethical concerns.

**Claims And Evidence:**

No

**Requested Changes:**

Some of the suggestions/requested changes are already mentioned in the weakness sections. Addition suggestions are below:

- In the second last paragraph of the paper, it says the “DermaMNIST, exhibited poor results” and attributes this to a possible reason of “ … sample size was insufficient, (Table 1.)”. For a binary class dataset, 10K samples may not be small. I would suggest adding either an experimental justification or/and a reference to existing work showing low samples being the cause for inferior self-supervised representation.

- In the last paragraph of the paper, it says the proposed augmented t-SimCNE “... can be useful for quality control, highlighting artefacts and problems in the data”. However, from the experiments included in the paper, it is unclear how it is useful in the above mentioned ways. Please clarify by referencing the results.

**Minor:**

- In Table 2, the bolding convention is difficult to understand, for example, for the t-SimCNE results for DermaMNIST, none of the numbers are bold, while the same for BloodMNIST two of the numbers are bold. It would be better to define the bolding convention in the caption explicitly. Same for tables 3 and 5.

- In Table 2, the caption says “PCam16 experiments had only one run due to its large size”, however, it is a sub-Imagenet size dataset (a standard for studying SSL models). For completeness, it would be beneficial to complete all three runs for PCam 16 with the variance reported in the table.

- In section 4, subsection “Evaluation in 2D”, the paper mentions “kNN classification accuracy (Pedregosa et al., 2011)”, however, the citation here, i.e. (Pedregosa et al., 2011) is for a Python machine learning library. For correctness, original papers should be cited, i.e. Fix, Evelyn; Hodges, Joseph L. (1951) and Cover, Thomas M.; Hart, Peter E. (1967), while the library should be explicitly/separately cited.

**Reference:**

Fix, Evelyn. Discriminatory analysis: nonparametric discrimination, consistency properties. Vol. 1. USAF school of Aviation Medicine, 1985.

Cover, Thomas, and Peter Hart. "Nearest neighbor pattern classification." IEEE transactions on information theory 13.1 (1967): 21-27.

**Strengths And Weaknesses:**

**Strengths:**

1. **Useful problem domain:** t-SimCNE for medical images provides evidence of practical application of the method which is useful for capturing informative 2D visualization of natural images to the medical microscopy images.

2. **Addition of logical augmentation:** Using 90-degree and random rotation seems to be logical for dermatology and Histopathological microscopy images, where there is no canonical orientation of the image and should be invariant to such rotations, Kang et al. (2023).

3. **Interesting t-SimCNE and t-SNE plots:** Figure 2, t-SimCNE vs t-SNE shows how a learnable visualization space can lead to better visualization of the data when compared to applying t-SNE to the pretrained features for Leukemia images.

___

**Weakness:**

1. **Title may overstate scope:** The paper title suggests an application and adaptation of t-SimCNE to a broader medical image domain, which sets the expectation that the method is extensively tested across diverse medical image datasets. However, the study focuses primarily on one dermatology dataset and four histopathology datasets. While these datasets are valuable, the title could better reflect the specific scope of the experiments presented in the paper.

2. **Limited new insights:** The main contribution of the paper is adding rotation augmentation to an existing t-SimCNE model. However, previous works, Kang et al. 2023 and Veeling et al. 2018, already mention that pathology images can be perturbed using random rotation, as finding a canonical orientation for these images is difficult, Kang et al. 2023. Simultaneously, the application of t-SimCNE to pathology and dermatological data by applying an existing work (t-SimCNE) to another domain (pathology images) provides limited new insights.

3. **Lack of Extensive Study in Pathology and Dermatology Domains:**

- The paper evaluates five datasets using two metrics, but the analysis appears limited. As seen in Table 2, for kNN accuracy with t-SimCNE, the 90-degree rotation (as in SimCLR, Chen et al. 2020) outperforms random rotation on two datasets, while random rotation performs better on only one. For the DermMNIST dataset, all augmentation methods show comparable results within the variance limits, and on one dataset, both 90-degree and random rotation perform equivalently.

- A similar pattern is observed for the Silhouette score with t-SimCNE, where predefined 90-degree rotations and random rotations outperform each other on only one, two, and two datasets, respectively. This makes it unclear when to use 90-degree rotations versus random rotations. While the paper offers subjective explanations for the underperformance of certain augmentations on specific datasets, these explanations are dataset-specific, and there is no experimental evidence to suggest whether these findings generalize to other pathology or dermatology datasets.

- To substantiate the use of random rotation augmentation in t-SimCNE, a more extensive evaluation across additional medical datasets is needed. A broader benchmark, such as MedMNIST v2, which covers 28 MNIST-like medical domain datasets, would provide stronger validation and more generalizable insights.

---

> ### Author Response · Authors · 2024-10-25
> **Response to Reviewer YPGk**
>
> We would like to thank the Reviewer for detailed and very useful feedback!
>
> > Title may overstate scope
>
> A similar issue was raised by another Reviewer. We have now changed the title to “Self-supervised Visualisation of Microscopy Datasets”.
>
> > Limited new insights: The main contribution of the paper is adding rotation augmentation to an existing t-SimCNE model. However, previous works, Kang et al. 2023 and Veeling et al. 2018, already mention that pathology images can be perturbed using random rotation, as finding a canonical orientation for these images is difficult, Kang et al. 2023. Simultaneously, the application of t-SimCNE to pathology and dermatological data by applying an existing work (t-SimCNE) to another domain (pathology images) provides limited new insights.
>
> Since the t-SimCNE method is relatively new and suggested a novel approach to self-supervised-based visualisations, we believe that applying it to a new domain (the original t-SimCNE paper only used CIFAR datasets) is an interesting contribution. We hope that our paper will draw the attention of the medical image analysis community towards self-supervised visualisations.
>
> > As seen in Table 2, for kNN accuracy with t-SimCNE, the 90-degree rotation (as in SimCLR, Chen et al. 2020) outperforms random rotation on two datasets, while random rotation performs better on only one. For the DermMNIST dataset, all augmentation methods show comparable results within the variance limits, and on one dataset, both 90-degree and random rotation perform equivalently.
> >
> > A similar pattern is observed for the Silhouette score with t-SimCNE, where predefined 90-degree rotations and random rotations outperform each other on only one, two, and two datasets, respectively. This makes it unclear when to use 90-degree rotations versus random rotations. While the paper offers subjective explanations for the underperformance of certain augmentations on specific datasets, these explanations are dataset-specific, and there is no experimental evidence to suggest whether these findings generalize to other pathology or dermatology datasets.
>
> In our opinion, the main take-home message is that using vertical flips and rotations on microscopy data helps self-supervised visualisations and self-supervised learning in general. This is in agreement with prior work that we cite. (To clarify: please note that Chen et al 2020 did *not* use 90-degree rotations, because they only worked with natural images). Whether to use arbitrary (non-90-degree) rotations or not, is less important and only had a minor effect on our results. Therefore we do not think it is a crucial point to investigate further.
>
> > To substantiate the use of random rotation augmentation in t-SimCNE, a more extensive evaluation across additional medical datasets is needed. A broader benchmark, such as MedMNIST v2, which covers 28 MNIST-like medical domain datasets, would provide stronger validation and more generalizable insights.
>
> We agree, and we added this statement more explicitly into the Discussion stating that “we found that blood microscopy datasets benefited the most from random rotations”, as an avenue for future work. Note that we did use MedMNIST v2, and in fact used almost all of its 2D microscopy datasets. Other modalities such as 3D or X-ray images are beyond the scope of our paper.
>
> > In the second last paragraph of the paper, it says the “DermaMNIST, exhibited poor results” and attributes this to a possible reason of “ … sample size was insufficient, (Table 1.)”. For a binary class dataset, 10K samples may not be small. I would suggest adding either an experimental justification or/and a reference to existing work showing low samples being the cause for inferior self-supervised representation.
>
> Indeed, that sentence in the discussion was formulated suboptimally. We do not think that poor results on DermaMNIST were *only* due to its sample size. Instead, the classes in this dataset show very little separation. This is in agreement with  supervised classification results reported in the literature: the MedMNIST v2 paper (https://www.nature.com/articles/s41597-022-01721-8/tables/4) reported 76.8% classification accuracy, which is close to the kNN accuracy in the embedding space that we obtained. Note that the majority class in DermaMNIST takes 67% of samples.
>
> We have reformulated that sentence in the Discussion.

---

> > ### Author Response · Authors · 2024-10-25
> >
> > > In the last paragraph of the paper, it says the proposed augmented t-SimCNE “... can be useful for quality control, highlighting artefacts and problems in the data”. However, from the experiments included in the paper, it is unclear how it is useful in the above mentioned ways. Please clarify by referencing the results.
> >
> > We thank the reviewer for raising this issue. To address this, we have now added two additional experiments shown in Fig. 6a and 6b in the revised manuscript. In the first experiment, we introduced 100 duplicate images with small variations into the dataset and trained the t-SimCNE embedding from scratch. In the resulting embedding, the duplicate images clustered together and were easy to detect (Fig. 6a). In the second experiment,  we introduced artefacts (black pixels) into 50 images. This had a similar effect: the visualization put the modified images into an artefact cluster. We believe that these additional experiments demonstrate that the 2D embeddings via t-SimCNE can be useful for quality control.
> >
> > > In Table 2, the bolding convention is difficult to understand, for example, for the t-SimCNE results for DermaMNIST, none of the numbers are bold, while the same for BloodMNIST two of the numbers are bold. It would be better to define the bolding convention in the caption explicitly. Same for tables 3 and 5.
> >
> > Thank you for pointing this out.  We added explanations to the table captions.  The bold numbers show which augmentation strategy (out of three different strategies) led to the best performance with t-SimCNE, and when the confidence intervals overlapped, we highlighted multiple values. For DermaMNIST we thought the results were not informative due to the very low overall performance, see above.
> >
> > > In Table 2, the caption says “PCam16 experiments had only one run due to its large size”, however, it is a sub-Imagenet size dataset (a standard for studying SSL models). For completeness, it would be beneficial to complete all three runs for PCam 16 with the variance reported in the table.
> >
> > Unfortunately, we did not have the time and computational resources to complete this during the rebuttal period, due to overlapping conference travels.  We will include additional runs into the final paper.
> >
> > > In section 4, subsection “Evaluation in 2D”, the paper mentions “kNN classification accuracy (Pedregosa et al., 2011)”, however, the citation here, i.e. (Pedregosa et al., 2011) is for a Python machine learning library. For correctness, original papers should be cited, i.e. Fix, Evelyn; Hodges, Joseph L. (1951) and Cover, Thomas M.; Hart, Peter E. (1967), while the library should be explicitly/separately cited.
> >
> > Thank you. We have added both references in the respective text.

---

### Decision · Action_Editor_oJNy · 2024-11-12

**Recommendation:** Reject

**Comment:**

Reviewers all agree about the decision. They found that:
- "the main take-home message is that using vertical flips and rotations on microscopy data helps self-supervised visualisations and self-supervised learning in general." In the prior case, there is a limited insight we can get from the paper. For the latter case, the paper needs restructuring, to analyze more diverse data distributions, specially natural image datasets. *Reviewer YPGk*
- Lack of baseline method, only t-SNE and variant of it is used as baseline, which cannot support the claim that t-SimCNE is a fit for medical image clustering. *Reviewer P8ZB*
- The datasets are very simila. Although there are differences in terms of cell appearance, they are still very similar and simple compared to common computer vision tasks. *Reviewer P8ZB*
- Most datasets include images of size 28x28, thus not being sufficiently large to make clustering a challenging task. *Reviewer P8ZB*
- The augmentations considered in the ablation study are limited, only crop, color jitter and grayscale are tested for their impact on clustering performance *Reviewer P8ZB*

Overall, the paper is in a preliminary exploratory phase. Potentially interesting findings could emerge by deepening the analysis. A resubmission is possible, provided that the major issues are addressed.

**Audience:**

Given the limited scope of the analysis, the results of the paper fail to meet the criteria of interest. Overall, it is unclear how generalizable the insights from the proposed data augmentation strategies are beyond the five similar datasets considered, and it is unlikely that the TMLR audience (or any subset of it) will find them particularly interesting. Important questions that must be addressed to improve the quality and generalizability of the work include: 1) When do the proposed augmentations work, and when do they not? 2) Why do they work, and why do they not?

**Claims And Evidence:**

The paper proposes applying an existing self-supervised visualization technique, called t-SimCNE, to microscopy datasets. The addition of data augmentation strategies, including vertical flips and rotations, is shown to improve visualization performance compared to vanilla self-supervised visualisation methods. Given the experimental nature of the paper, a more thorough analysis would be expected. All reviewers agree that the experiments are limited in both scale and scope, and as a result, their insights lack sufficient depth to ensure generalizability.

**Resubmission Of Major Revision:**

The authors may consider submitting a major revision at a later time.